# Cervical spine immobilisation following blunt trauma in pre-hospital and emergency care: A systematic review

Abdullah Pandor[1]*, Munira Essat[1], Anthea Sutton[1], Gordon Fuller[1], Stuart Reid[2], Jason E. Smith[3], Rachael Fothergill[4], Dhushy Surendra Kumar[5], Angelos Kolias[6], Peter Hutchinson[6], Gavin D. Perkins[7], Mark H. Wilson[8], Fiona Lecky[1]

1 SCHARR, University of Sheffield, Sheffield, United Kingdom, 2 Sheffield Teaching Hospitals NHS Foundation Trust, Sheffield, United Kingdom, 3 Department of Emergency, University Hospitals Plymouth NHS Trust, Plymouth, United Kingdom, 4 London Ambulance Service NHS Trust, London, United Kingdom, 5 Department of Critical Care, University Hospitals Coventry and Warwickshire, Coventry, United Kingdom, 6 Department of Clinical Neurosciences, Addenbrooke's Hospital & University of Cambridge, Cambridge, United Kingdom, 7 Warwick Medical School, University of Warwick, Coventry, United Kingdom, 8 Imperial College London, St Mary's Hospital, London, United Kingdom

* a.pandor@sheffield.ac.uk

**Data Availability Statement:** All relevant data are within the manuscript and its Supporting Information files.

## Abstract

### Objectives

To assess whether different cervical spine immobilisation strategies (full immobilisation, movement minimisation or no immobilisation), impact neurological and/or other outcomes for patients with suspected cervical spinal injury in the pre-hospital and emergency department setting.

### Design

Systematic review following Preferred Reporting Items for Systematic Reviews and Meta-Analyses guidelines.

### Data sources

MEDLINE, EMBASE, CINAHL, Cochrane Library and two research registers were searched until September 2023.

### Eligibility criteria

All comparative studies (prospective or retrospective) that examined the potential benefits and/or harms of immobilisation practices during pre-hospital and emergency care of patients with a potential cervical spine injury (pre-imaging) following blunt trauma.

### Data extraction and synthesis

Two authors independently selected and extracted data. Risk of bias was appraised using the Cochrane ROBINS-I tool for non-randomised studies. Data were synthesised without meta-analysis.

**Funding:** This study was funded by the United Kingdom National Institute for Health and Care Research (NIHR) Health Technology Assessment (HTA) Programme (project number 131430, https://fundingawards.nihr.ac.uk/award/ NIHR131430). The funder provided support in the form of salaries (paid to institutions) for all authors but did not have any additional role in the study design, data collection and analysis, decision to publish, or preparation of the manuscript. The specific roles of these authors are articulated in the 'author contributions' section. The views expressed in this paper are those of the authors and not necessarily those of the NIHR or the Department of Health and Social Care. Any errors are the responsibility of the authors.

**Competing interests:** All authors declare grant funding to their employing institutions from the United Kingdom National Institute for Health and Care Research (NIHR) Health Technology Assessment (HTA) Programme, as outlined in the funding statement. These competing interests do not alter our adherence to PLOS ONE policies on sharing data and materials.

## Results

Six observational studies met the inclusion criteria. The methodological quality was variable, with most studies having serious or critical risk of bias. The effect of cervical spine immobilisation practices such as full immobilisation or movement minimisation during pre-hospital and emergency care did not show clear evidence of benefit for the prevention of neurological deterioration, spinal injuries and death compared with no immobilisation. However, increased pain, discomfort and anatomical complications were associated with collar application during immobilisation.

## Conclusions

Despite the limited evidence, weak designs and limited generalisability, the available data suggest that pre-hospital cervical spine immobilisation (full immobilisation or movement minimisation) was of uncertain value due to the lack of demonstrable benefit and may lead to potential complications and adverse outcomes. High-quality randomised comparative studies are required to address this important question.

## Trial registration

**PROSPERO REGISTRATION** Fiona Lecky, Abdullah Pandor, Munira Essat, Anthea Sutton, Carl Marincowitz, Gordon Fuller, Stuart Reid, Jason Smith. A systematic review of cervical spine immobilisation following blunt trauma in pre-hospital and emergency care. PROSPERO 2022 CRD42022349600 Available from: https://www.crd.york.ac.uk/prospero/ display_record.php?ID=CRD42022349600.

## Introduction

Spinal cord injury is a life-changing event usually caused by road traffic crashes, sports injury or falls [1]. Though rare, the annual incidence rate varies globally between regions and countries, ranging from 3.6 to 195.4 cases per million population worldwide [2]. Evidence reviews indicate that almost half of all traumatic spinal cord injuries involve the cervical spine, and usually result from cervical spine dislocation [1]. Cervical spinal cord injury is associated with high mortality rates, healthcare and societal costs [3]. Survivors usually experience significant disability, and a major impact on their (and their carers') quality of life.

Pre-hospital spinal immobilisation has been a component of "potential spinal injury" standard of care for over 20 years [4]. When a patient is thought to have a potential cervical spine injury, current practice by pre-hospital emergency medical service clinicians varies but the default will often be to place the patient on a rigid transportation device and immobilise the neck using tape supported by head-blocks and a semi-rigid cervical collar. This full 'triple' immobilisation in theory reduces movement and aims to prevent more damage to the spine during transfer to hospital and in the Emergency Department (ED) prior to imaging. The key concept driving spinal immobilisation was a perception that neurological deterioration (increasing weakness of muscles and/or loss of sensation) after spinal injury resulted from a failure to properly immobilise the patient's spine [5]. As such, immobilisation of the cervical spine on the slightest suspicion of injury is generally recommended in current guidelines [1,6],

trauma courses and has been adopted worldwide by many pre-hospital emergency medical services [7–9].

Despite this practice, the biomechanical evidence that unrestricted patient movement can cause or worsen spinal cord injury is lacking and it is now believed likely that when neurological deterioration occurs it primarily results from swelling and bleeding occurring within the fixed space of the spinal canal [10]. There is also unease that certain patient groups, for example those with ankylosing spondylitis, may suffer neurological deterioration as a result of neck positioning in a rigid collar [11]. It is now thought possible that full cervical spinal immobilisation during pre-hospital and emergency care may cause more harm than benefit, by increasing work of breathing, risk of aspiration (inhaling blood, vomit or other secretions) and development of pressure sores or by worsening concomitant brain injury (through increasing intracranial pressure) [12–15]. Cervical spine immobilisation undoubtedly increases pain and discomfort after injury and is not tolerated by people with pre-existing cognitive impairment, agitation after injury and distressed children [1,16]. Furthermore meticulous attention to cervical spine management may distract clinicians from treating immediately life-threatening traumatic pathology and prolong on-scene times. This evolving understanding has led some emergency medical services to abandon all or part of cervical spine immobilisation [17], but there is variation in practice resulting from uncertainty as to its benefits [1,12]. Hence, with this systematic review we aimed to determine if selection of different cervical spine immobilisation practices—during the pre-hospital and emergency department care of patients with possible cervical spine injury—impacts neurological and other outcomes.

## Methods

A systematic review was undertaken in accordance with the general principles recommended in the Preferred Reporting Items for Systematic Reviews and Meta-Analyses (PRISMA) statement [18] and was registered on the International Prospective Register of Systematic Reviews (PROSPERO) database (CRD42022349600) [19].

### Eligibility criteria

All comparative studies (prospective or retrospective) evaluating spinal immobilisation strategies during pre-hospital or emergency care were eligible for inclusion. The study population of interest in our review consisted of all pre-hospital and ED patients (any age) with the potential for cervical spine injury (pre-imaging). The systematic review sought to elicit comparative patient and healthcare outcomes associated with different prehospital and emergency department cervical spine immobilisation strategies. Therefore, we excluded studies utilising healthy human volunteers (non-trauma), cadaver or manikin models where units of movement or force are often reported but not their effect on clinical outcomes. Studies reporting solely on operative spinal stabilisation, strategies for selecting patients for spinal imaging, helmet removal techniques, degree of spinal movement during emergency intubation and use of spinal orthoses not used in emergency care were also excluded. We limited the review to patients injured through blunt trauma (i.e. road traffic collisions, sport injuries, falls and other blunt mechanisms) and excluded studies of penetrating injuries as these present much less frequently, were not the patient group of interest, and their anatomical injury patterns are fundamentally different to those with blunt trauma. It was also noted that a previous systematic review reported evidence of patient harm when penetrating neck injuries are immobilised [14]. Full cervical spinal immobilisation was defined as the use of hard/semi-rigid cervical collar with head blocks and tape/strap fixation to underlying surface (triple immobilisation) applied by pre-hospital or ED clinicians as per ATLS protocols [9]. We considered

comparative cervical spine immobilisation strategies less than triple immobilisation including movement minimisation (defined as using any single or combination of two triple immobilisation elements) and no immobilisation (using no elements of triple immobilisation). The key outcomes of interest were spinal neurological deterioration (as determined by any motor and/or sensory deficit that appeared or worsened after contact with emergency medical services and that persisted until discharge), spinal injury presence and/or severity, complications potentially related to collar use (e.g., aspiration, pressure sores, raised intracranial pressure), brain injury, critical care and hospital stay, mortality and acceptability to patients and paramedics.

## Data sources and searches

Potentially relevant studies were identified through searches of several electronic databases and research registers. This included MEDLINE (OvidSP from 1946), EMBASE (OvidSP from 1974), CINAHL (EBSCO from 1981), the Cochrane Library (https://www.cochranelibrary.com from inception), ClinicalTrials.gov (US National Institutes of Health from 2000) and the International Clinical Trials Registry Platform (World Health Organisation from 1990). The search strategy used free text and thesaurus terms and combined synonyms relating to the topic of interest (e.g. spinal injury and immobilisation strategies) with pre-hospital or emergency care terms. No language or date restrictions were used. However, as the current review updated the search strategy of two broadly overlapping previous systematic reviews [13,15], searches were limited by date from 2015 (last search date from earlier reviews) [13] to September 2023. Searches were supplemented by hand-searching the reference lists of all relevant studies (including existing systematic reviews); forward citation searching of included studies (via Web of Science, formerly known as Web of Knowledge); contacting key experts in the field; and undertaking targeted searches of the World Wide Web using the Google search engine. Further details of the search strategies can be found in **S1 Table**.

## Study selection

All titles were examined for inclusion by one reviewer (ME) and any citations that clearly did not meet the inclusion criteria (e.g. non-human, unrelated to spinal immobilisation) were excluded. All abstracts and full text articles were then examined independently by two reviewers (ME and AP). For quality assurance purposes, all excluded citations, abstracts and full text studies were independently checked by clinical experts (FL and GF). Any disagreements in the selection process were resolved through discussion with the wider group (SR, and JS) and included by consensus.

## Data extraction and quality assessment

For eligible studies, data relating to study design, methodological quality and outcomes were extracted by one reviewer (ME) into a standardised data extraction form and independently checked for accuracy by a second reviewer (AP). Any discrepancies were resolved through discussion, or if this was unsuccessful, wider group opinion was sought (FL, GF, SR). Where multiple publications of the same study were identified, data were extracted and reported as a single study.

The methodological quality of each included study was assessed using a risk of bias tool (RoB) recommended by the Cochrane Collaboration: ROBINS-I for non-randomised interventions studies [20]. In general, this tool determines the RoB in various domains, including confounding bias, selection bias (participant selection), misclassification bias (classification of interventions), performance bias (deviations from the intended interventions) attrition bias

(missing outcome data), detection bias (measurement of the outcome) and reporting bias (selection of the reported result). Every domain includes a series of signalling questions to help the assessors judge the RoB, based on the responses given to the signalling questions. The overall domain-level judgement within the ROBINS-I tool, was rated as low, moderate, serious, or critical. An overall RoB for each study was defined as low risk when all domains were judged as low; moderate risk when all domains were judged as low or moderate RoB; serious risk when one or more domains were considered as serious but not at critical RoB in any domain and critical risk when at least one domain was judged to be at critical RoB.

### Data synthesis and analysis

Due to significant levels of heterogeneity between studies (study design, participants, inclusion criteria) and variable reporting of items, a meta-analysis was not considered possible. As a result, a pre-specified narrative synthesis approach [21] was undertaken, with data being summarised in tables with accompanying narrative summaries that included a description of the included variables, statistical methods and effect estimates, where applicable. ROB figures were generated using *robvis* software [22]. We initially examined cervical spinal immobilisation strategies that compared triple immobilisation with any cervical management less than triple immobilisation (i.e. no immobilisation/movement minimisation). However, due to the emerging variation in the methods used to immobilise trauma patients in pre-hospital or emergency care [1,12] and to fully explore cervical immobilisation strategies we also evaluated movement minimisation strategies when compared to no immobilisation (a post hoc change).

### Patient and public involvement

This review was conducted as part of a feasibility assessment of the Spinal Injury Study (SIS: randomised controlled trial). Although patients and the public were not specifically involved in the design or conduct of this systematic review, they were represented in the SIS investigator team.

## Results

### Study flow

**Fig 1** summarises the process of identifying and selecting relevant literature. Of the 1811 citations identified, 6 studies met the inclusion criteria. Of these, only two published studies [23,24] and one unpublished study by Thompson L, Shaw G, McMeekin P, Hawkins C, McClelland G. [Unpublished] compared full cervical spinal immobilisation with any cervical management less than triple immobilisation (i.e. no immobilisation/movement minimisation). The remaining studies [25–27] compared movement minimisation with no immobilisation. Sixty full-text articles were excluded as they did not meet all the pre-specified inclusion criteria. The majority of the articles were excluded primarily on the basis of an inappropriate study design (i.e. non comparative study), or wrong target population (i.e. not cervical spine immobilisation following blunt trauma in pre-hospital and emergency care). More specifically, one potentially relevant paper [28] was excluded due the lack information on cervical spinal immobilisation strategies in the published paper (including the failure of the study authors to provide the requested details). Another potentially relevant study [29] was excluded as it only focused on patients with on-scene cardiac arrest owing to blunt trauma and did not provide relevant outcome data for patients with suspected cervical spinal injury. A full list of excluded studies with reasons for exclusion is provided in **S2 Table**.

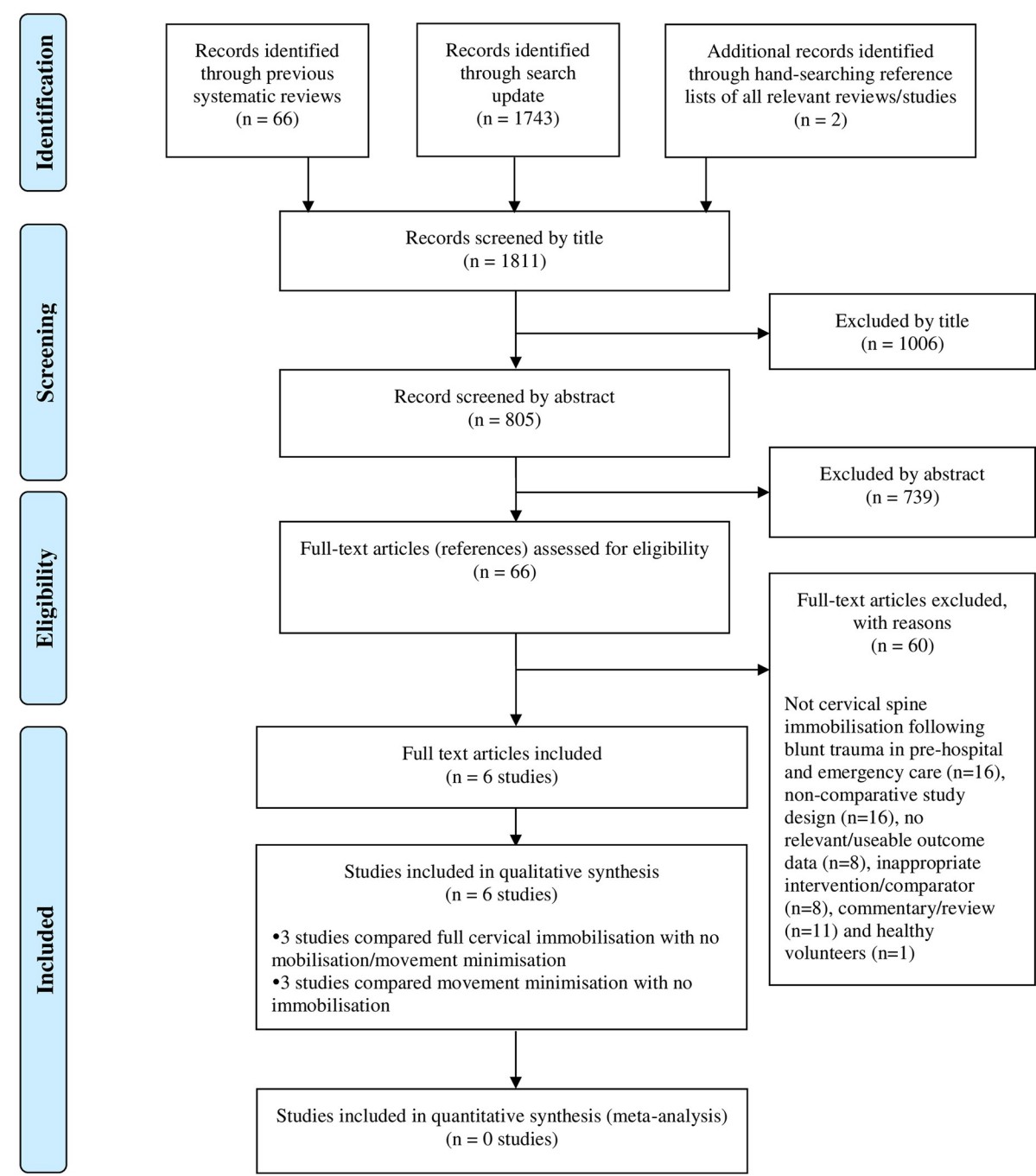

**Fig 1. Study flow chart (adapted).**

## Study and patient characteristics

The design and participant characteristics of the 6 included studies are summarised in **Table 1**. Apart from one unpublished UK study (a summary study report was provided on request), all other studies were published between 1998 and 2021 and were undertaken in Australia (n = 1) [25], Taiwan (n = 1) [27], the USA (n = 2) [24,26] and one study was in multiple locations [23]. Sample sizes varied widely and ranged from 56 to 5139 [27] participants and the mean age ranged from 35 [23] to 78.2 [24] years (not reported in 2 studies) [25,26].

**Table 1. Study and population characteristics.**

| Author, year | Country (setting) | Design | Single/ multi-centre | Sample size | Population | Period | Mean age (years) | Female | Mechanism (and type) of injury | Intervention | Comparator | Primary outcome(s) |
|---|---|---|---|---|---|---|---|---|---|---|---|---|
| **Full 'triple' immobilisation vs. no immobilisation** | | | | | | | | | | | | |
| Hauswald et al. 1998 [23] | Malaysia, USA (pre-hospital) | Retrospective analysis (chart review) | Multi (2 sites) | 454 | Patients with acute blunt traumatic spinal or spinal cord injuries | January 1988 to January 1993 | 35 | 20.3% | Falls: 28%; RTA: 65%; Other: 7% (100% blunt) | Full spinal immobilisation,[a] (USA cohort, n = 334) | No spinal immobilisation (Malaysia cohort, n = 120) | Rate of neurologic injury |
| **Full 'triple' immobilisation vs. movement minimisation** | | | | | | | | | | | | |
| Thompson et al. [Unpublished] [b] | UK (pre-hospital) | Prospective controlled "before after" interventional study [c] | Single | 56 | Trauma patients (≥18 years) with suspected cervical spine injury | December 2020 to August 2021 | 62.3 | 50.0% | Falls: 68%; RTA: 23%; Other: 9% (NR) [d] | Full spinal immobilisation (defined as the use of semi-rigid collar, blocks, and tape +/- orthopaedic stretcher/scoop or vacuum mattress; n = 30) | Movement minimisation (defined as the use of blocks and tape but no semi-rigid collar (n = 26) | Time: on scene, to imaging, in ED and new neurology |
| Underbrink et al. 2018 [24] | USA (pre-hospital) | Retrospective before-and-after study | Multi (9 sites) | 237 | Adults (≥60 years) with a cervical spine injury (fracture or cord) | January 2012 and June 2014 to July 2014 and December 2015 | 78.2 | 48.5% | Falls: 65%; RTA: 23%; Other: 12% (NR) [d] | Full spinal immobilisation (defined as the use of backboard, cervical collar and head immobilisation devices; Before cohort, n = 123) | Movement minimisation (defined as the use of collar only; After cohort, n = 114) | NR but included immobilisation type, presence of neurological deficit, patient disposition at discharge, and in-hospital mortality/ hospice |
| **Movement minimisation vs. No immobilisation** | | | | | | | | | | | | |
| Asha et al. 2021 [25] | Australia (pre-hospital and ED) | Retrospective analysis (chart review) | Multi (7 sites) | 2036 | Patients with suspected traumatic cervical spine injury | October 2017 to July 2018 | NR (median, 54) | 44.1% | Falls: 39%; RTA: 24% (motorcycle: 6%; motor vehicle: 18%); Other: 37% (NR) [d] | Movement minimisation (defined as [1] the pre-hospital and ED use of hard collar until imaging and then removed if no injury identified (n = 268) or [2] pre-hospital hard collar, and then soft collar in ED until imaging. If injured changed to hard collar or removed if no injury identified, n = 1133) | No immobilisation (n = 582) | Proportion who developed new or worsening neurological deficit |

*(Continued)*

**Table 1.** (Continued)

| Author, year | Country (setting) | Design | Single/multi-centre | Sample size | Population | Period | Mean age (years) | Female | Mechanism (and type) of injury | Intervention | Comparator | Primary outcome(s) |
|---|---|---|---|---|---|---|---|---|---|---|---|---|
| Leonard et al. 2012 [26] | USA (pre-hospital and ED) | Prospective cohort study | Single | 285 | Children (<18 years) with suspected traumatic cervical spine injury | July 2003 to August 2004 | NR | 47.0% | Falls: 29%; RTA: 43%; Other: 28% (NR) [d] | Movement minimisation (defined as the use of cervical collar and/or rigid spine board, n = 173) | No immobilisation (n = 112) | Level of pain on ED arrival and rate of cervical spine imaging |
| Lin et al. 2011 [27] | Taiwan (pre-hospital) | Retrospective analysis (chart review) | Single | 5139 | Patients who sustained lightweight motorcycle (engine size <150 mL) injuries, assumed to have been at a low velocity (<50 km/h), with suspected cervical spine injury | January 2008 to December 2009 | 38 | 45.1% | RTA: 100% (motorcycle) (100% blunt) | Movement minimisation (defined as the use of cervical collar brace only, n = 2605) | No immobilisation (n = 2534) | Incidence of cervical spine injury |

Abbreviations: ED, emergency department; NR, not reported; RTA, road traffic accident.

[a] Not clearly defined. We assumed this based on description in the introductory text which states 'Immobilization is improved by using a firm surface; addition of a hard cervical collar, head blocks, and lateral restraint provides progressively more stability. . . Patients are fully immobilized at the injury site if there is any suggestion that the neck or back could be injured. Immobilization is usually continued in the ED until the spine is "cleared" by multiple imaging procedures.

[b] For further details see https://www.neas.nhs.uk/our-services/research-and-development/smrf.aspx.

[c] This study was originally designed as a feasibility randomised controlled trial (https://www.isrctn.com/ISRCTN11400471); however, due to the impact and restrictions of the COVID-19 pandemic and limited resources intervention/control assignment was made based on two three-month timeframe periods into before (full immobilisation) and after (movement minimisation) groups.

[d] Although not explicitly reported in the published manuscript, we assumed that most patients had blunt trauma based on the mechanism of injury.

| Author, year | Risk of Bias Domains | | | | | | | |
|---|---|---|---|---|---|---|---|---|
| | Confounding | Participant selection | Classification of interventions | Deviations of intended interventions | Missing data | Measurement of outcomes | Selection of reported results | Overall |
| Asha et al. 2021 [25] | ✗ | − | − | ? | − | − | − | Serious |
| Hauswald et al. 1998 [23] | ✗ | ! | − | ? | ✗ | + | − | Critical |
| Leonard et al. 2012 [26] | ✗ | ✗ | + | ? | ? | − | − | Serious |
| Lin et al. 2011 [27] | ! | ! | − | ? | ✗ | − | − | Critical |
| Thompson et al. [Unpublished] | + | + | + | + | + | ✗ | − | Serious |
| Underbrink et al. 2018 [24] | ✗ | ! | − | ? | ✗ | − | − | Critical |

⊕ indicates low risk of bias; ⊖ indicates moderate risk of bias; ✗ indicates serious risk of bias; ! indicates critical risk of bias and ? indicates no information

**Fig 2. Summary of each study's risk of bias using the ROBINS-I tool–review authors' judgements.**

## Risk of bias assessments

The overall methodological quality of the 6 included studies is summarised in **Figs 2 and 3** (further details of the review author judgements is provided in **S3 Table**).

The methodological quality of the included studies was variable, with most studies having serious or critical risk of bias in at least one item of the ROBINS-I tool. The main risk of bias limitations were related to bias due to confounding (with the exception of one unpublished study, none of the other studies used an appropriate analysis method to control for all important confounders and studies remained at risk of residual or unmeasured confounding) [23–27], patient selection factors (arising from retrospective data collection [23–25,27], and selected populations) [23,24,27], bias due to missing data (most of the studies provided details on the proportions of missing data across intervention groups, reasons for missing data, and how missing data was handled) and unblinded measurement of outcomes. Of note, the three key studies [23,24,27] at critical risk of bias had only studied patients with documented spinal or cervical spinal injury as opposed to the whole population who had received either immobilisation strategy.

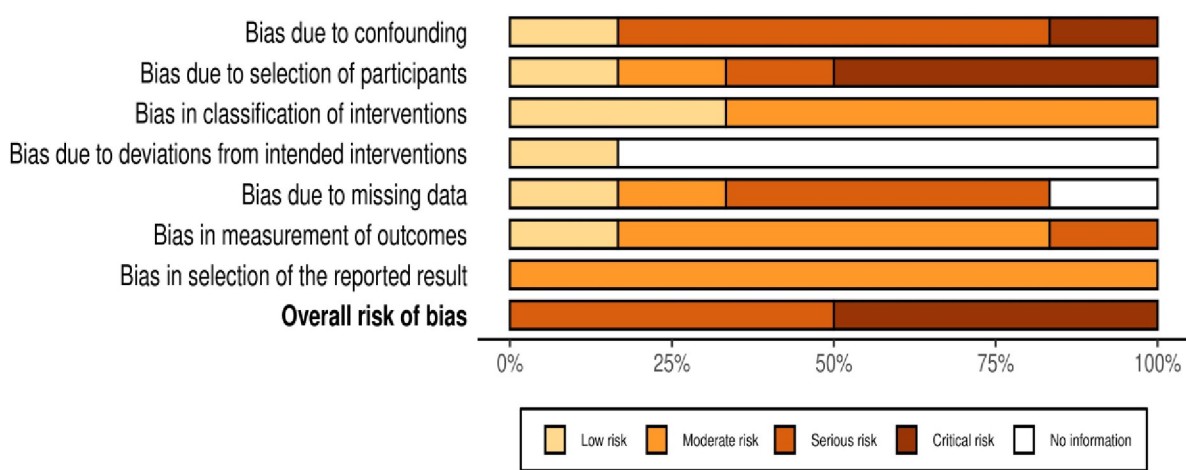

**Fig 3. Risk of bias assessment summary graph using the ROBINS-I tool—review authors' judgements.**

## Effect of interventions (summary of results)

**Neurological deterioration.** Two studies provided information on the effect of cervical spinal immobilisation on neurological deterioration (Table 2A). Both studies demonstrated no difference in the rates of neurological deterioration with differing cervical spine immobilisation strategies in patients with suspected blunt cervical spinal trauma. In the prospective before-and-after study by Thompson et al., [Unpublished] there was no neurological deterioration in either group (full immobilisation versus movement minimisation in 56 patients). In contrast, in the larger retrospective cohort study of movement minimisation versus no immobilisation in 2036 patients [25], 9 patients neurologically deteriorated after ED arrival in the movement minimisation group compared to zero in the no immobilisation group. However, the groups were not comparable in terms of confounders at baseline on ED arrival. Neurological deficit on ED arrival (due to brain or spinal cord injury) was almost twice as common in the movement minimisation group. Furthermore, neurological deterioration could only be attributed to spinal injury in two out of nine people in which it developed.

**Spinal/Neurological injuries.** Six studies examined the effect of cervical spinal immobilisation on spinal injuries and/or spinal cord injury (Table 2b). Thompson et al., [Unpublished] found no statistically significant difference between full 'triple' immobilisation compared with movement minimisation strategies on rates of spinal injury (odds ratio [OR] = 6.75, 95% confidence interval [CI]: 0.33–136.96; p = 0.21). Two retrospective chart review studies, reported by Asha et al., [25] and Lin et al., [27] observed no potential benefit of maintaining cervical spine immobilisation through movement minimisation strategies compared with no immobilisation in adults (OR = 4.11, 95% CI: 2.13–7.95; p<0.0001 and OR = 4.20, 95% CI: 2.23–7.89; p < .00001, respectively). In general, higher rates of cervical spine injury were observed in the movement minimisation groups compared with the no immobilisation groups. One prospective cohort study [26] highlighted the rarity of cervical spine injuries in children, finding only one cervical spine injury within a cohort of 285 children (OR = 1.96, 95% CI: 0.08–48.45; p = 0.68).

Two retrospective studies solely included people with documented spinal injury and studied rates of neurological/ spinal cord injury. In a before-and-after study, Underbrink et al., [24] found no differences in the rates of cervical spinal cord injury between older adult people with cervical spine injury who either received full triple immobilisation or movement minimisation strategies (OR = 1.25, 95% CI: 0.42–3.73; p = 0.69). Similarly, Hauswald et al., [23] found no differences in the rates of neurological disability between two transnational cohorts with isolated cervical spine injury who had received full triple immobilisation as compared to no immobilisation (OR = 1.29, 95% CI: 0.57–2.93; p = 0.54).

**Complications and/or other outcomes.** Only a limited number of studies provided information on the complications of cervical spine immobilisation. Based on a qualitative experience survey of 18 adult participants (full 'triple' immobilisation, n = 14; movement minimisation without collar, n = 4), conducted alongside a prospective controlled before-and-after study, Thompson et al., [Unpublished] observed increased pain/discomfort (which also contributed to negative experiences which may have further limited its utility e.g. excessive pain and anatomical issues which may limit collar application) with semi-rigid collar application during full immobilisation although there were limited data from the non-collar group to allow meaningful comparisons between groups. In this study, there were no reported cases of immobilisation induced pressure damage in either the intervention or control groups. Leonard et al., [26] found that children suspected of traumatic cervical spine injury who were immobilised through movement minimisation (i.e., having cervical collar in place and or being secured to a rigid spine board) reported increased pain scores compared with non-spine

**Table 2. Summary of neurological and spinal injury outcomes in patients with suspected traumatic cervical spine injury.** a) Neurological deterioration. b) Spinal/neurological injury.

| Author, year | Outcomes | Definition | Findings | | Effect estimate (author calculated) |
|---|---|---|---|---|---|
| | | | Intervention (n/N) | Control (n/N) | |
| Thompson et al. [Unpublished] | Neurological deterioration | New neurology was assessed on clinical examination where there was any compromise to sensory, motor or reflex function or presence of neurogenic shock. This followed the criteria highlighted by the International Standards for Classification of Spinal Cord Injury [30] | Full immobilisation: 0/30 (0%) | Movement minimisation: 0/26 (0%) | OR: not estimable |
| Asha et al. 2021 [25] | Neurologic deficit present on arrival to ED | NR | Movement minimisation–Combined: 92/1401 (6.6%) [a] | No immobilisation: 21/582 (3.6%) | OR: 1.88 (95% CI: 1.16, 3.05; p = 0.01) |
| | New neurologic deficit during hospital stay | The development of neurological deficit during the hospital admission was determined from the discharge summary but included none organic cause (all imaging normal and neurologic deficit resolved while in hospital), progression of intracranial injuries, and spinal cord injury | Movement minimisation–Combined: 9/1401 (0.6%) [b] | No immobilisation: 0/582 (0%) | OR: 7.95 (95% CI: 0.46, 136.78; p = 0.15) |

| Author, year | Outcomes | Definition | Findings | | Effect estimate (author calculated) |
|---|---|---|---|---|---|
| | | | Intervention (n/N) | Control (n/N) | |
| Thompson et al. [Unpublished] | Cervical spine injury | NR | Full spinal immobilisation: 3/30 (10%) | Movement minimisation: 0/26 (0%) | OR: 6.75 (95% CI: 0.33; 136.96; p = 0.21) |
| Underbrink et al. 2018 [24] | Cervical cord injury | NR (used ICD-9 diagnosis codes to identify patients with cervical cord injury) | Full spinal immobilisation: 8/123 (6.5%) | Movement minimisation: 6/114 (5.3%) | OR: 1.25 (95% CI: 0.42, 3.73; p = 0.69) |
| Asha et al. 2021 [25] | Cervical spine injury | Cervical spine cord injury defined as any changes (based on imaging) consistent with an acute injury such as fractures, dislocations, ligamentous disruptions and spinal cord trauma | Movement minimisation–Combined: 94/1401 (6.7%) [a] | No immobilisation: 10/582 (1.7%) | OR: 4.11 (95% CI: 2.13, 7.95; p<0.0001) |
| Leonard et al. 2012 [26] | Cervical spine injury | NR | Movement minimisation–Combined: 1/173 (0.6%) | No immobilisation: 0/112 (0%) | OR: 1.96 (95% CI: 0.08, 48.45; p = 0.68) |
| Lin et al. 2011 [27] | Cervical spine injury | Cervical spine injury defined as any recorded change in neurologic status, including bony lesion of cervical spine or spinal cord injury, visualised on CT or magnetic resonance imaging | Movement minimisation: 51/2605 (2.0%) | No immobilisation: 12/2534 (0.5%) | OR: 4.20 (95% CI: 2.23; 7.89; p<0.00001) [b] |

(Continued)

**Table 2.** (Continued)

| | | Full spinal immobilisation: 70/334 (21%) | No immobilisation: 13/120 (11%) | OR: 2.18 (95% CI: 1.16, 4.11; p = 0.02) c |
|---|---|---|---|---|
| Hauswald et al. 1998 [23] | Neurological disability: all patients with spinal immobilisation (cervical, thoracic and lumbosacral injury) | | | |
| | Disability defined as complete quadriplegia or paraplegia, inability to ambulate without assistance, incontinence, or the need for chronic catheterisation, and those who died | | | |
| | Neurological disability: all patients with isolated cervical injury only | Full spinal immobilisation: 34/113 (30%) | No immobilisation: 10/40 (25%) | OR: 1.29 (95% CI: 0.57, 2.93; p = 0.54) d |
| | Disability defined as complete quadriplegia or paraplegia, inability to ambulate without assistance, incontinence, or the need for chronic catheterisation, and those who died | | | |

Abbreviations: ED, emergency department; NR, not reported; OR, odds ratio.

[a] Movement minimisation—Rigid collar group: 29/268 (10.8%); Movement minimisation—Soft collar group: 63/1133 (5.6%).

[b] Movement minimisation—Rigid collar group: 3/268 (1.1%); Movement minimisation—Soft collar group: 6/1133 (0.5%). Of the 9 with neurological deterioration 3 had no organic cause and resolved during admission, 3 were due to progression of brain injury and 2 attributable to spinal cord injury and one of uncertain aetiology.

Abbreviations: ICD-9, International Classification of Diseases 9th revision; NR, not reported; OR, odds ratio.

[a] Movement minimisation—Rigid collar group: 17/268 (6.3%); Movement minimisation—Soft collar group: 77/1133 (6.8%).

[b] Additional information reported in paper–no significant correlation observed when comparing cervical spine injury patients with movement minimisation and no immobilisation ($\chi^2$, p = 0.896).

[c] Information reported in paper—adjusted (for age, sex, level of injury, and mechanism of injury) OR: 2.03; 95% CI: 1.03, 3.99; p = 0.04.

[d] Information reported in paper—adjusted (for age, sex, level of injury, and mechanism of injury) OR: 1.52; 95% CI 0.64, 3.62; p = 0.34.

immobilised children (OR = 2.2; 95% CI: 1.4–3.4; p<0.05) and were more likely to undergo cervical radiography (56.6% versus 13.4%, respectively; p<0.0001) and be admitted to hospital (41.6% versus 14.3%, respectively; p<0.05). However, both groups had comparable median lengths of stay in the ED (2.8 versus 2.8 hours, respectively; p>0.05). In contrast, for patients involved in lightweight motorcycle injuries sustained at low speed (<50 km/h) and who sustained a cervical spine injury, Lin et al., [27] reported longer intensive care unit stays for those immobilised through movement minimisation compared to those with no immobilisation (7.54 ±7.93 versus 2.33 ±1.63 days, respectively; p = 0.002), whereas no difference was found in the total length of hospital stay (17.43 ±9.35 versus 12.00 ±8.89 days, respectively; p = 0.154).

Three studies reported data on mortality. Underbrink et al., [24] found there was no significant difference (after adjusting for injury severity) in hospital mortality among those who were fully immobilised compared with movement minimisation strategies (19.5% versus 9.7% respectively; adjusted OR = 0.56, 95% CI: 0.24–1.30; p = 0.18). Similarly, in studies comparing movement minimisation strategies with no immobilisation, Asha et al., [25] (2.7% (combined groups) versus 2.2%, respectively; OR = 1.22, 95% CI: 0.65–2.31, p = 0.54) and Lin et al., [27] (no deaths in both groups) also found no significant differences in hospital deaths.

Only one study evaluated compliance with immobilisation. In this study by Thompson et al., [Unpublished] where all participants were managed by pre-hospital clinicians, 17% (n = 5) of participants were not compliant with having a semi-rigid collar applied when being fully immobilised (n = 30), whereas all participants (n = 26) were compliant with movement minimisation (no semi-rigid collar applied). As noted by the authors, non-compliance with immobilisation originated from complications of semi-rigid collar use and was attributed to excessive pain (n = 3) and due to complications arising from patient anatomy (n = 2) making collar application impractical in this study.

## Discussion

### Summary of results

This systematic review identified 6 comparative studies that examined the potential benefits and/or harms of cervical spine immobilisation practices during pre-hospital and emergency care in patients with a potential cervical spine injury (pre-imaging) following blunt trauma. Despite the limited comparative evidence, substantial risk of bias concerns, and limited generalisability, the available data suggest that cervical spine immobilisation practices such as full immobilisation or movement minimisation during pre-hospital and emergency care may not have clear benefit for the prevention of neurological deterioration, spinal injuries and death compared with no immobilisation. However, there are concerns of increased pain, discomfort and anatomical complications associated with collar application during immobilisation.

### Interpretation of results

Despite widespread dissemination of guidelines and varied observed approaches to patient immobilisation, the practice of universal pre-hospital spinal immobilisation (full immobilisation or movement minimisation) in patients with blunt trauma to the cervical spine is supported by very limited robust evidence of effectiveness. None of six observational studies (two prospective and four retrospective—8207 participants in total) reported any improved outcome associated with a greater degree of cervical spine immobilisation. Five studies including an unpublished study reported no difference in comparative rates of cervical spinal cord injury [23,24,26,27] whilst the sixth–at high risk of confounding—reported a significantly higher rate when comparing movement minimisation to no immobilisation [25]. However, in the context of uncontrolled heterogeneous studies, we do acknowledge that absence of evidence is not

evidence of no benefit. There was limited reporting of comparative complications; however, each of the two prospectively recruiting studies found that greater cervical spine immobilisation was associated with higher levels of pain and discomfort. Thompson et al., [Unpublished] also found that as well as distressing immobilised patients this limits adherence, and can also consume scarce ED staff resource—through need for repeated reassurance and/ or prescription and administration of analgesia.

Studies and expert consensus continue to emerge questioning the use of pre-hospital spinal immobilisation. Some regions have now moved away from universal cervical spine immobilisation in conscious, fully alert, stable and co-operative patients [17]. As suggested by Hauswald et al. [23] a significant amount of force is required to create an unstable injury to the cervical spine during the initial trauma, and additional movements of the spine are unlikely to cause further damage. However, for high-risk patients unable to protect their own cervical spine (e.g. those with a reduced level of consciousness, or apparently under the influence of alcohol and/or drugs) a policy of immobilisation remains in place [17,31] with the intention of preventing the devastating effects of cord damage.

Only one of the identified publications studied paediatric cervical spine immobilisation [26]. As there are anatomical differences between adults and children this may prevent valid generalisations of the adult literature to spinal immobilisation in a paediatric population. Similarly, only one study [24] focused specifically on the elderly (aged ≥60 years) with a cervical spine injury (spinal fracture or cord injury). Given the increasing volume of trauma in the frail and the elderly, referred to as the 'silver trauma tsunami' [32] current practice needs to take account of the challenges of immobilisation in this older cohort [33].

## Comparison to the existing literature

Several broadly overlapping systematic reviews have been published that have investigated pre-hospital spinal immobilisation [13–15,34,35]. Their findings are broadly similar to the current review, with widely heterogeneous results and high risk of bias in the majority of included studies. Each note the lack of robust evidence to support cervical spine immobilisation and significant potential for harm. Unlike these systematic reviews [13–15,34,35], and others [36,37], our review included only comparative studies, and specifically excluded studies which were conducted in healthy human volunteers (non-trauma), cadaver or manikin models, and patients with penetrating neck or spinal trauma. The relevance and generalisability of the inclusion of these studies to injured humans who have a suspected blunt spinal injury would have been questionable due to the absence of pain, difficulty in replication of biomechanical behaviour of an injured spine in the field, and the lack of evidence that parameters of force or movement will translate to clinical outcomes.

To distinguish from emergency immobilisation, a systematic review by Brannigan et al., [38] based on limited and low-quality evidence, also found that prolonged use of hard collars (≥2 days of wear) was associated with significant morbidity including pressure sores, dysphagia, raised intracranial pressure and peripheral nerve palsies and may be detrimental to a patient's outcome.

## Strengths and limitations

This systematic review has several strengths. It is the first systematic review to evaluate cervical spine immobilisation practices (full immobilisation, movement minimisation or no immobilisation) during pre-hospital and emergency department care in patients (any age) with a potential cervical spine injury (pre-imaging) following blunt trauma. The review was conducted with robust methodology in accordance with the PRISMA statement [18] and the protocol was

registered with the PROSPERO register. Clinical experts, in addition to the core review team, were involved and consulted throughout as advisors and to assess the validity and applicability of research findings during the review processes. We also used the ROBINS-I tool, which addresses RoB against an absolute scale rather than using the ideal observational study as a standard [20]. As such, this is the only tool recommended by the Cochrane Collaboration for assessing RoB in non-randomised interventions studies and has been widely adopted for use in published and planned systematic reviews [39].

The main limitations of this study relate to the constraints of the studies reviewed and their own limitations. Most of the included studies used retrospective designs [23–25,27] with limitations on data quality, confounding issues and failure (and consistency between studies) to accurately ascertain outcomes. Conversely, better quality data may be obtained with prospective cohorts, but these studies were limited by smaller sample sizes and lack statistical power.

The presence of substantial levels of heterogeneity between studies precluded any meta-analysis and or statistical examination of the causes of heterogeneity due to the small number of studies. Potential sources of heterogeneity included variation in study design, the study population, injury mechanisms, interventions and outcome definition and measurement. As a result, we reported descriptive statistics to provide a better understanding of the evidence base applicable to the subject matter, and shortcomings regarding reliability and validity of the data.

Given the paucity of evidence it is possible that we may have missed studies that were neither registered nor published, resulting in publication bias. In addition, we did not perform hand searching (i.e., manual page-by-page examination of the entire contents) of journals or conference proceedings and did not include regional bibliographic databases, although the yield of such searches is generally low [40]. Finally, decisions on study relevance, information gathering, and validity of articles were unblinded and could potentially have been influenced by pre-formed opinions. However, blinding researchers is resource intensive with uncertain benefits in protecting against bias decisions [41].

## Implications for policy, practice and future research

It remains standard practice in the UK and worldwide for patients with suspected cervical spine injury in pre-hospital and ED settings to have their cervical spine immobilised [1,6–9]. However, in recent years there has been intense debate in the pre-hospital and emergency medicine community as to whether and how the cervical spine should be routinely immobilised in patients with suspected cervical spine injury. The medical and legal concern of missing a cervical spinal injury has lent strong support for this 'extraordinarily conservative' approach of liberal pre-hospital immobilisation to almost all trauma patients with suspected injury [31]. Clinician concerns persist around the possibility of contributing to or causing a catastrophic neurological decline in a patient with a neck injury. There is also increased awareness amongst clinicians of the potential side-effects of immobilising the cervical spine. Our review provides no clear robust evidence of any protective benefit in routinely immobilising the cervical spine in patients with suspected cervical spine injuries. Clinicians should be aware of this lack of evidence to support the routine and liberal use of cervical spine immobilisation; their clinical decision-making tools should acknowledge this and potential for adverse consequences. However, prior to amending existing guidance further research is essential to provide clear evidence-based criteria in applying cervical spine immobilisation. To this end, we currently await the results of the ongoing National Institute of Health and Care Research (NIHR) Spinal Immobilisation Study (SIS), a multi-centre, open-label, pragmatic, pre-hospital, non-inferiority randomised controlled trial evaluating the effectiveness of immobilisation regimes

involving neck movement minimisation and standard triple immobilisation (current NHS practice) in patients with potential or suspected cervical spine injury following blunt trauma recruited in a pre-hospital (ambulance) setting. This study is due to complete in October 2025 [42].

## Conclusions

Current practice of routine cervical spine immobilisation for patients injured by blunt trauma appears outdated; our review has shown that pre-hospital and emergency department pre-imaging immobilisation of the cervical spine is of uncertain value due to the lack of demonstrated benefit and potential complications associated with it use. We recommend further high-quality randomised comparative studies to inform evidence-based guidance for emergency immobilisation of the cervical spine.

## Supporting information

**S1 Table. Literature search strategies.**
(DOCX)

**S2 Table. List of excluded studies with rationale.**
(DOCX)

**S3 Table. Summary of each study's risk of bias using the ROBINS-I tool–review authors' judgements in detail.**
(DOCX)

## Acknowledgments

The authors would like to thank all additional members of the core project group for NIHR131430 for input and commentary throughout the work. We are also indebted to Joanne Hinde for assistance with logistics and administration.

## Author Contributions

**Conceptualization:** Abdullah Pandor, Munira Essat, Anthea Sutton, Gordon Fuller, Stuart Reid, Jason E. Smith, Rachael Fothergill, Dhushy Surendra Kumar, Angelos Kolias, Peter Hutchinson, Gavin D. Perkins, Mark H. Wilson, Fiona Lecky.

**Data curation:** Abdullah Pandor, Munira Essat, Anthea Sutton.

**Formal analysis:** Abdullah Pandor, Munira Essat, Gordon Fuller, Stuart Reid, Fiona Lecky.

**Funding acquisition:** Abdullah Pandor, Munira Essat, Anthea Sutton, Gordon Fuller, Stuart Reid, Jason E. Smith, Rachael Fothergill, Dhushy Surendra Kumar, Angelos Kolias, Peter Hutchinson, Gavin D. Perkins, Mark H. Wilson, Fiona Lecky.

**Methodology:** Abdullah Pandor, Munira Essat, Anthea Sutton, Gordon Fuller, Stuart Reid, Jason E. Smith, Rachael Fothergill, Dhushy Surendra Kumar, Angelos Kolias, Peter Hutchinson, Gavin D. Perkins, Mark H. Wilson, Fiona Lecky.

**Project administration:** Abdullah Pandor, Fiona Lecky.

**Validation:** Abdullah Pandor, Munira Essat, Gordon Fuller, Fiona Lecky.

**Writing – original draft:** Abdullah Pandor, Munira Essat, Gordon Fuller, Stuart Reid, Fiona Lecky.

**Writing – review & editing:** Abdullah Pandor, Munira Essat, Anthea Sutton, Gordon Fuller, Stuart Reid, Jason E. Smith, Rachael Fothergill, Dhushy Surendra Kumar, Angelos Kolias, Peter Hutchinson, Gavin D. Perkins, Mark H. Wilson, Fiona Lecky.

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
