## [Decision Letter · Decision Letter 0]

12 Dec 2023

PONE-D-23-31240Cervical spine immobilisation following blunt trauma in pre-hospital and emergency care: a systematic reviewPLOS ONE

Dear Dr. Pandor,

Thank you for submitting your manuscript to PLOS ONE. After careful consideration, we feel that it has merit but does not fully meet PLOS ONE’s publication criteria as it currently stands. Therefore, we invite you to submit a revised version of the manuscript that addresses the points raised during the review process.

We look forward to receiving your revised manuscript.

Kind regards,

Alaa Oteir, PhD

Academic Editor

PLOS ONE

Journal Requirements:

The authors would like to thank all additional members of the core project group for NIHR131430 for input and commentary throughout the work. We are also indebted to Joanne Hinde for assistance with logistics and administration. 

This study was funded by the United Kingdom National Institute for Health and Care Research Health Technology Assessment Programme (project number 131430).  The views expressed in this paper are those of the authors and not necessarily those of the NIHR or the Department of Health and Social Care. Any errors are the responsibility of the authors. The funders had no role in the study design, in the collection, analysis and interpretation of data; in the writing of the manuscript; and in the decision to submit the manuscript for publication.

AP, GF, AS, ME, JS, DSK, MW, GP, PJ, AH, AK, RF and FL all declare funding from NIHR Health Technology Assessment Grant NIHR131430 – Spinal Injury Study. There are no other competing interests.

We note that one or more of the authors are employed by a commercial company: NIHR Health Technology. 

“The funder provided support in the form of salaries for authors, but did not have any additional role in the study design, data collection and analysis, decision to publish, or preparation of the manuscript. The specific roles of these authors are articulated in the ‘author contributions’ section.”

4. We note that you have referenced Thompson L, C B, Shaw G, McMeekin P, Hawkins C, McClelland G. which has currently not yet been accepted for publication. Please remove this from your References and amend this to state in the body of your manuscript: Thompson L, C B, Shaw G, McMeekin P, Hawkins C, McClelland G. [Unpublished]”) as detailed online in our guide for authors

Reviewers' comments:

Reviewer's Responses to Questions

**Comments to the Author**

1. Is the manuscript technically sound, and do the data support the conclusions?

Reviewer #1: Yes

Reviewer #2: Yes

2. Has the statistical analysis been performed appropriately and rigorously? 

Reviewer #1: N/A

Reviewer #2: Yes

3. Have the authors made all data underlying the findings in their manuscript fully available?

Reviewer #1: Yes

Reviewer #2: Yes

4. Is the manuscript presented in an intelligible fashion and written in standard English?

Reviewer #1: Yes

Reviewer #2: Yes

5. Review Comments to the Author

Reviewer #1: This study considers an important and current issue. The study is well conceived, delivered and presented. The data is presented in a logical and accessible fashion. The prose is well written and accessible.

One area of feedback relates to clarity in relation to the search strategy and the link to the PROSPERO registration.

- Please confirm if you searched the International Clinical Trials Registry Platform (World Health Organisation from 1990) – I don’t see this in the PROSPERO registration or the supplemental files but in the manuscript

- Please can you adjust figure 1 to demonstrate where the “additional records found through other sources” (what was the source / which part of the search strategy were these generated by e.g. contacting key experts in the field; and undertaking targeted searches of the World Wide Web using the Google search engine

Please ensure that critical risk of bias is easily distinguishable from serious risk of bias in all figures.

Reviewer #2: The objective of this systematic review was to evaluate the impact of different cervical spine immobilization strategies (full immobilization, movement minimization, or no immobilization) on neurological and other outcomes for patients with suspected cervical spinal injury in pre-hospital and emergency department settings. The review included comparative studies examining the benefits and harms of immobilization practices during pre-hospital and emergency care for patients with potential cervical spine injuries following blunt trauma. Six observational studies met the inclusion criteria, but methodological quality varied, with most studies having a serious or critical risk of bias. The evidence did not clearly support the benefit of cervical spine immobilization practices in preventing neurological deterioration, spinal injuries, or death compared to no immobilization. However, collar application during immobilization was associated with increased pain, discomfort, and anatomical complications. The authors concluded that despite limited evidence and weak study designs, pre-hospital cervical spine immobilization may have uncertain value due to a lack of demonstrable benefit and potential complications. They emphasized the need for high-quality comparative studies to address this important question. Indeed, this topic is interesting; however, there are some comments and concerns that should be addressed.

Introduction

-While the introduction is detailed, it could benefit from more conciseness. Some sentences are lengthy and complex, making it challenging for the reader to follow the main points.

-Consider breaking down complex sentences into more digestible segments for improved clarity.

-Ensure consistent citation style throughout the introduction. Some citations are provided in square brackets, while others are in regular parentheses. Consistency in citation style is essential for academic writing.

-The introduction occasionally uses informal language, such as "devastating consequence" and "a devastating impact." While these phrases convey the severity of the issue, consider using more neutral and formal language to maintain an academic tone.

-The transition from discussing the global incidence of spinal cord injuries to the specific focus on cervical spine injuries could be smoother. Consider providing a clearer bridge between these topics for a more seamless flow.

-Some details, such as the specific equipment used in pre-hospital spinal immobilization, may be more detailed than necessary for the introduction. Consider focusing on essential information to maintain reader engagement and relevance.

-The introduction effectively leads to the study's aim, but the sentence introducing the aim is complex. Consider simplifying the sentence structure to clearly convey the research question and objective.

-The discussion on the evolving understanding of spinal immobilization is informative, but there is some redundancy in emphasizing the potential harm without introducing new information. Streamline the content to avoid repetition and maintain reader engagement.

-Review the text for minor grammatical issues, such as subject-verb agreement and punctuation, to enhance the overall grammatical accuracy.

Methods

The methods section is well-structured and adheres to the standard guidelines for systematic reviews. However, a few suggestions for improvement can be considered:

- While the eligibility criteria are generally well-defined, there could be additional clarification on why certain exclusions were made, especially regarding the exclusion of studies involving healthy human volunteers, cadaver or manikin models. Providing a brief rationale for these exclusions would enhance the transparency of the selection process.

- The definition of full cervical spinal immobilization is clear. However, the description of cervical spine management less than triple immobilization is somewhat intricate. Consider simplifying the language for better comprehension, ensuring that readers can easily understand the criteria for studies falling into this category.

Discussion

- The summary of results is clear and concise. However, it might be beneficial to present some key quantitative data, such as the number of studies supporting or refuting the effectiveness of cervical spine immobilization. Providing a brief overview of the characteristics of the included studies (e.g., study design, sample size) could enhance the reader's understanding.

- While the discussion mentions concerns about pain, discomfort, and anatomical complications associated with collar application during immobilization, consider providing more context on the potential clinical significance of these complications. Discussing the clinical implications will help readers better understand the practical implications of the findings.

- The comparison to existing literature is informative, but consider discussing any novel or divergent findings in more detail. Highlighting areas where the current review differs from previous ones will contribute to a more nuanced understanding of the research landscape.

- The strengths and limitations are appropriately discussed. To further enhance transparency, consider providing more details on the rationale behind the selection of the risk of bias tool (ROBINS-I) and why it was deemed appropriate for this review.

- The implications for policy and practice are well-discussed. However, consider providing more details on the potential impact of the review findings on clinical guidelines and decision-making in emergency care. Additionally, while the ongoing NIHR Spinal Immobilisation Study is mentioned, consider briefly discussing how its results might influence future practice.

- The conclusions are clear and aligned with the study findings. To strengthen the conclusion, consider reiterating the key takeaway messages for clinicians and policymakers. Emphasize the need for evidence-based criteria in applying cervical spine immobilization.

6. PLOS authors have the option to publish the peer review history of their article (what does this mean?). If published, this will include your full peer review and any attached files.

Reviewer #1: No

Reviewer #2: No

---

## [Author Response · Author response to Decision Letter 0]

30 Jan 2024

Comments Authors’ response

Journal requirements 

Please ensure that your manuscript meets PLOS ONE's style requirements, including those for file naming… 

 We can confirm that the manuscript meets PLOS ONE’s style requirements.

The authors would like to thank all additional members of the core project group for NIHR131430 for input and commentary throughout the work. We are also indebted to Joanne Hinde for assistance with logistics and administration. 

This study was funded by the United Kingdom National Institute for Health and Care Research Health Technology Assessment Programme (project number 131430). The views expressed in this paper are those of the authors and not necessarily those of the NIHR or the Department of Health and Social Care. Any errors are the responsibility of the authors. The funders had no role in the study design, in the collection, analysis and interpretation of data; in the writing of the manuscript; and in the decision to submit the manuscript for publication.

 We have updated the Acknowledgements Section and Funding Statement accordingly and added it at the bottom of the cover letter.

AP, GF, AS, ME, JS, DSK, MW, GP, PJ, AH, AK, RF and FL all declare funding from NIHR Health Technology Assessment Grant NIHR131430 – Spinal Injury Study. There are no other competing interests.

We note that one or more of the authors are employed by a commercial company: NIHR Health Technology. 

“The funder provided support in the form of salaries for authors, but did not have any additional role in the study design, data collection and analysis, decision to publish, or preparation of the manuscript. The specific roles of these authors are articulated in the ‘author contributions’ section.”

 The National Institute for Health and Care Research (NIHR) is a research funding body in the UK and is funded by the UK Department of Health and Social Care to improve the health and wealth of the nation through research. The Health 

 Technology Assessment (HTA) Programme is the largest of the NIHR programmes.

 We have updated the Funding and Competing Interests Statements accordingly and added it at the bottom of the cover letter. 

4. We note that you have referenced Thompson L, C B, Shaw G, McMeekin P, Hawkins C, McClelland G. which has currently not yet been accepted for publication. Please remove this from your References and amend this to state in the body of your manuscript: Thompson L, C B, Shaw G, McMeekin P, Hawkins C, McClelland G. [Unpublished]”) as detailed online in our guide for authors

 We have updated the reference style as suggested.

 We have updated the Supporting Information style as suggested.

Reviewers' comments 

Reviewer #1

Reviewer #1: This study considers an important and current issue. The study is well conceived, delivered and presented. The data is presented in a logical and accessible fashion. The prose is well written and accessible. 

 Thank you for your in-depth review and your comments on the work.

 We agree this topic is important, very relevant and timely.

 In response to your suggestions and questions, see our responses below.

One area of feedback relates to clarity in relation to the search strategy and the link to the PROSPERO registration.

- Please confirm if you searched the International Clinical Trials Registry Platform (World Health Organisation from 1990) – I don’t see this in the PROSPERO registration or the supplemental files but in the manuscript

 To clarify, we searched the International Clinical Trials Registry Platform as a grey literature source to identify ongoing and unpublished studies. We believe this is covered in our protocol by the statement "…systematic searches of study 

 registries…".

- Please can you adjust figure 1 to demonstrate where the “additional records found through other sources” (what was the source / which part of the search strategy were these generated by e.g. contacting key experts in the field; and undertaking targeted searches of the World Wide Web using the Google search engine

 Thank you, we have amended Figure 1 accordingly.

Please ensure that critical risk of bias is easily distinguishable from serious risk of bias in all figures. 

 The risk of bias figures were generated using the robvis software, as noted in our methods section. As we are unable to edit the robvis outputs, we have updated the figures using an alternate output format provided by the robvis software 

 (i.e. colorblind-friendly scheme).

Reviewer #2

The objective of this systematic review was to evaluate the impact of different cervical spine immobilization strategies (full immobilization, movement minimization, or no immobilization) on neurological and other outcomes for patients with suspected cervical spinal injury in pre-hospital and emergency department settings. The review included comparative studies examining the benefits and harms of immobilization practices during pre-hospital and emergency care for patients with potential cervical spine injuries following blunt trauma. Six observational studies met the inclusion criteria, but methodological quality varied, with most studies having a serious or critical risk of bias. The evidence did not clearly support the benefit of cervical spine immobilization practices in preventing neurological deterioration, spinal injuries, or death compared to no immobilization. However, collar application during immobilization was associated with increased pain, discomfort, and anatomical complications. The authors concluded that despite limited evidence and weak study designs, pre-hospital cervical spine immobilization may have uncertain value due to a lack of demonstrable benefit and potential complications. They emphasized the need for high-quality comparative studies to address this important question. Indeed, this topic is interesting; however, there are some comments and concerns that should be addressed.

 Thank you for your in-depth review, and your positive comments on the conduct, standard and writing of our work. 

 In response to your suggestions and questions, please see our responses below.

 We agree this topic is important, very relevant and timely.

Introduction

-While the introduction is detailed, it could benefit from more conciseness. Some sentences are lengthy and complex, making it challenging for the reader to follow the main points.

-Consider breaking down complex sentences into more digestible segments for improved clarity.

 We have shortened sentences as advised for clarity. 

-Ensure consistent citation style throughout the introduction. Some citations are provided in square brackets, while others are in regular parentheses. Consistency in citation style is essential for academic writing.

 We can confirm that the manuscript meets PLOS ONE’s style requirements

-The introduction occasionally uses informal language, such as "devastating consequence" and "a devastating impact." While these phrases convey the severity of the issue, consider using more neutral and formal language to maintain an academic tone. 

 We have replaced these terms with “life changing” and “major”.

-The transition from discussing the global incidence of spinal cord injuries to the specific focus on cervical spine injuries could be smoother. Consider providing a clearer bridge between these topics for a more seamless flow. 

 We have replaced this with “Though rare, the annual incidence rate varies globally between regions and countries, ranging from 3.6 to 195.4 cases per million population worldwide [2]. Evidence reviews indicate that almost half of all 

 traumatic spinal cord injuries involve the cervical spine, and usually result from cervical spine dislocation [1].“ 

-Some details, such as the specific equipment used in pre-hospital spinal immobilization, may be more detailed than necessary for the introduction. Consider focusing on essential information to maintain reader engagement and relevance.

-The introduction effectively leads to the study's aim, but the sentence introducing the aim is complex. Consider simplifying the sentence structure to clearly convey the research question and objective. 

 We have removed some equipment detail whilst retaining the description of triple immobilisation recommended in current guidance. 

 We have simplified this sentence “Hence, with this systematic review we aimed to determine if selection of different cervical spine immobilisation practices - during the pre-hospital and emergency department care of patients with possible 

 cervical spine injury - impacts neurological and other outcomes.” 

-The discussion on the evolving understanding of spinal immobilization is informative, but there is some redundancy in emphasizing the potential harm without introducing new information. Streamline the content to avoid repetition and maintain reader engagement.

 Thank you, we thought it important to detail potential harms to contextualise both current variation in practice and selection of systematic review outcomes.

-Review the text for minor grammatical issues, such as subject-verb agreement and punctuation, to enhance the overall grammatical accuracy. 

 This has been done.

Methods

The methods section is well-structured and adheres to the standard guidelines for systematic reviews. However, a few suggestions for improvement can be considered:

 Thank you, please see our responses below.

- While the eligibility criteria are generally well-defined, there could be additional clarification on why certain exclusions were made, especially regarding the exclusion of studies involving healthy human volunteers, cadaver or manikin models. Providing a brief rationale for these exclusions would enhance the transparency of the selection process.

 We have now added the following text to in eligibility criteria. “The systematic review sought to elicit comparative patient and healthcare outcomes associated with different prehospital and emergency department cervical spine immobilisation 

 strategies. Therefore we excluded studies utilising healthy human volunteers (non-trauma), cadaver or manikin models where units of movement or force are often reported but not their effect on clinical outcomes.”

- The definition of full cervical spinal immobilization is clear. However, the description of cervical spine management less than triple immobilization is somewhat intricate. Consider simplifying the language for better comprehension, ensuring that readers can easily understand the criteria for studies falling into this category

 We have attempted to simplify this language whilst preserving accuracy.

Discussion

- The summary of results is clear and concise. However, it might be beneficial to present some key quantitative data, such as the number of studies supporting or refuting the effectiveness of cervical spine immobilization. Providing a brief overview of the characteristics of the included studies (e.g., study design, sample size) could enhance the reader's understanding.

 Thank you, we have added the following text to “interpretation of findings”

 ”None of six observational studies (two prospective and four retrospective - 8207 participants in total) reported any improved outcome associated with a greater degree of cervical spine immobilisation. Five studies reported no difference in 

 comparative rates of cervical spinal cord injury [23, 24, 26, 27] whilst the sixth – at high risk of confounding - reported a significantly higher rate when comparing movement minimisation to no immobilisation [25]. However in the context of 

 uncontrolled heterogeneous studies, we do acknowledge that absence of evidence is not evidence of no benefit.”

- While the discussion mentions concerns about pain, discomfort, and anatomical complications associated with collar application during immobilization, consider providing more context on the potential clinical significance of these complications. Discussing the clinical implications will help readers better understand the practical implications of the findings.

 We have added the following text to interpretation of findings. “There was limited reporting of comparative complications; however, each of the two prospectively recruiting studies found that greater cervical spine immobilisation was 

 associated with higher levels of pain and discomfort. Thompson et al., [Unpublished] also found that as well as distressing immobilised patients this limits adherence, and can also consume scarce ED staff resource - through need for repeated 

 reassurance and/ or prescription and administration of analgesia.”

- The comparison to existing literature is informative, but consider discussing any novel or divergent findings in more detail. Highlighting areas where the current review differs from previous on

---

## [Decision Letter · Decision Letter 1]

28 Mar 2024

Cervical spine immobilisation following blunt trauma in pre-hospital and emergency care: a systematic review

PONE-D-23-31240R1

Dear Dr. Pandor

We’re pleased to inform you that your manuscript has been judged scientifically suitable for publication and will be formally accepted for publication once it meets all outstanding technical requirements.

Kind regards,

Alaa Oteir, PhD

Academic Editor

PLOS ONE

Additional Editor Comments (optional):

Reviewers' comments:

Reviewer's Responses to Questions

**Comments to the Author**

1. If the authors have adequately addressed your comments raised in a previous round of review and you feel that this manuscript is now acceptable for publication, you may indicate that here to bypass the “Comments to the Author” section, enter your conflict of interest statement in the “Confidential to Editor” section, and submit your "Accept" recommendation.

Reviewer #1: All comments have been addressed

Reviewer #2: All comments have been addressed

2. Is the manuscript technically sound, and do the data support the conclusions?

Reviewer #1: Yes

Reviewer #2: Yes

3. Has the statistical analysis been performed appropriately and rigorously? 

Reviewer #1: N/A

Reviewer #2: Yes

4. Have the authors made all data underlying the findings in their manuscript fully available?

Reviewer #1: Yes

Reviewer #2: Yes

5. Is the manuscript presented in an intelligible fashion and written in standard English?

Reviewer #1: Yes

Reviewer #2: Yes

6. Review Comments to the Author

Reviewer #1: (No Response)

Reviewer #2: I would like to thank the authors for this satisfactory modifications. Thus, I have no more comments and recommend accepting this paper.

7. PLOS authors have the option to publish the peer review history of their article (what does this mean?). If published, this will include your full peer review and any attached files.

Reviewer #1: No

Reviewer #2: No
